# NeurWIN: Neural Whittle Index Network for Restless Bandits via Deep RL

## Abstract

Whittle index policy is a powerful tool to obtain asymptotically optimal solutions for the notoriously intractable problem of restless bandits. However, finding the Whittle indices remains a difficult problem for many practical restless bandits with convoluted transition kernels. This paper proposes NeurWIN, a neural Whittle index network that seeks to learn the Whittle indices for any restless bandits by leveraging mathematical properties of the Whittle indices. We show that a neural network that produces the Whittle index is also one that produces the optimal control for a set of Markov decision problems. This property motivates using deep reinforcement learning for the training of NeurWIN. We demonstrate the utility of NeurWIN by evaluating its performance for three recently studied restless bandit problems. Our experiment results show that the performance of NeurWIN is either better than, or as good as, state-of-the-art policies for all three problems.

## 1 Introduction

Many sequential decision problems can be modeled as multi-armed bandit problems. A bandit problem models each potential decision as an arm. In each round, we play $M$ arms out of a total of $N$ arms by choosing the corresponding decisions. We then receive a reward from the played arms. The goal is to maximize the long-term total discounted reward. Consider, for example, displaying advertisements on an online platform with the goal to maximize the long-term discounted click-through rates. This can be modeled as a bandit problem where each arm is a piece of advertisement and we choose which advertisements to be displayed every time a particular user visits the platform. It should be noted that the reward, i.e., click-through rate, of an arm is not stationary, but depends on our actions in the past. For example, a user that just clicked on a particular advertisement may be much less likely to click on the same advertisement in the near future. Such a problem is a classic case of the *restless bandit problem*, where the reward distribution of an arm depends on its state, which changes over time based on our past actions.

The restless bandit problem is notoriously intractable (Papadimitriou & Tsitsiklis, 1999). Most recent efforts, such as *recovering bandits* (Pike-Burke & Grunewalder, 2019), *rotting bandits* (Seznec et al., 2020), and *Brownian bandits* (Slivkins & Upfal, 2008), only study some special instances of the restless bandit problem. The fundamental challenge of the restless bandit problem lies in the explosion of state space, as the state of the entire system is the Cartesian product of the states of individual arms. A powerful tool to address the explosion of state space is the Whittle index policy (Whittle, 1988). In a nutshell, the Whittle index policy calculates a Whittle index for each arm based on the arm's current state, where the index loosely corresponds to the amount of cost that we are willing to pay to play the arm, and then plays the arm with the highest index. It has been shown that the Whittle index policy is either optimal or asymptotically optimal in many settings.

In this paper, we present Neural Whittle Index Network (NeurWIN), a principled machine learning approach that finds the Whittle indices for virtually all restless bandit problems. We note that the Whittle index is an artificial construct that cannot be directly measured. Finding the Whittle index is typically intractable. As a result, the Whittle indices of many practical problems remain unknown except for a few special cases.

We are able to circumvent the challenges of finding the Whittle indices by leveraging an important mathematical property of the Whittle index: Consider an alternative problem where there is only one arm and we decide whether to play the arm in each time instance. In this problem, we need to pay a

constant cost of $\lambda$ every time we play the arm. The goal is to maximize the long-term discounted net reward, defined as the difference between the rewards we obtain from the arm and the costs we pay to play it. Then, the optimal policy is to play the arm whenever the Whittle index becomes larger than $\lambda$. Based on this property, a neural network that produces the Whittle index can be viewed as one that finds the optimal policy for the alternative problem for any $\lambda$.

Using this observation, we propose a deep reinforcement learning method to train NeurWIN. To demonstrate the power of NeurWIN, we employ NeurWIN for three recently studied restless bandit problems, namely, recovering bandit (Pike-Burke & Grunewalder, 2019), wireless scheduling (Aalto et al., 2015), and stochastic deadline scheduling (Yu et al., 2018). There is no known Whittle index for the first problem, and there is only an approximation of the Whittle index under some relaxations for the second problem. Only the third problem has a precise characterization of the Whittle index. For the first two problems, the index policy using our NeurWIN achieves better performance than existing studies. For the third problem, the index policy using our NeurWIN has virtually the same performance as the Whittle index policy.

The rest of the paper is organized as follows: Section 2 reviews related literature. Section 3 provides formal definitions of the Whittle index and our problem statement. Section 4 introduces our training algorithm for NeurWIN. Section 5 demonstrates the utility of NeurWIN by evaluating its performance under three recently studied restless bandit problems. Finally, Section 6 concludes the paper.

## 2 RELATED WORK

Restless bandit problems were first introduced in (Whittle, 1988). They are known to be intractable, and are in general PSPACE hard (Papadimitriou & Tsitsiklis, 1999). As a result, many studies focus on finding the Whittle index policy for restless bandit problems, such as in (Le Ny et al., 2008; Meshram et al., 2018; Tripathi & Modiano, 2019; Dance & Silander, 2015). However, these studies are only able to find the Whittle indices under various specific assumptions about the bandit problems.

There has been a lot of studies on applying RL methods for bandit problems. (Dann et al., 2017) proposed a tool called Uniform-PAC for contextual bandits. (Zanette & Brunskill, 2018) described a framework-agnostic approach towards guaranteeing RL algorithms' performance. (Jiang et al., 2017) introduced contextual decision processes (CDPs) that encompass contextual bandits for RL exploration with function approximation. (Riquelme et al., 2018) compared deep neural networks with Bayesian linear regression against other posterior sampling methods. However, none of these studies are applicable to restless bandits, where the state of an arm can change over time.

Deep RL algorithms have been utilized in problems that resemble restless bandit problems, including HVAC control (Wei et al., 2017), cyber-physical systems (Leong et al., 2020), and dynamic multi-channel access (Wang et al., 2018). In all these cases, a major limitation for deep RL is scalability. As the state spaces grows exponentially with the number of arms, these studies can only be applied to small-scale systems, and their evaluations are limited to cases when there are at most 5 zones, 6 sensors, and 8 channels, respectively.

An emerging research direction is applying machine learning algorithms to learn Whittle indices. (Borkar & Chadha, 2018) proposed employing the LSPE(0) algorithm (Yu & Bertsekas, 2009) coupled with a polynomial function approximator. The approach was applied in (Avrachenkov & Borkar, 2019) for scheduling web crawlers. However, this work can only be applied to restless bandits whose states can be represented by a single number, and it only uses a polynomial function approximator, which may have low representational power (Sutton & Barto, 2018). (Fu et al., 2019) proposed a Q-learning based heuristic to find Whittle indices. However, as shown in its experiment results, the heuristic may not produce Whittle indices even when the training converges.

## 3 PROBLEM SETTING

In this section, we provide a brief overview of restless bandit problems and the Whittle index. We then formally define the problem statement.

### 3.1 RESTLESS BANDIT PROBLEMS

A restless bandit problem consists of $N$ restless arms. In each round $t$, a control policy observes the state of each arm $i$, denoted by $s_i[t]$, and selects $M$ arms to activate. We call the selected arms as *active* and the others as *passive*. We use $a_i[t]$ to denote the policy's decision on each arm $i$, where $a_i[t] = 1$ if the arm is active and $a_i[t] = 0$ if it is passive at round $t$. Each arm $i$ generates a stochastic reward $r_i[t]$ with distribution $R_{i,act}(s_i[t])$ if it is active, and with distribution $R_{i,pass}(s_i[t])$ if it is passive. The state of each arm $i$ in the next round evolves by the transition kernel of either $P_{i,act}(s_i[t])$ or $P_{i,pass}(s_i[t])$, depending on whether the arm is active. The goal of the control policy is to maximize the total discounted reward, which can be expressed as $\sum_{t=1}^{\infty} \sum_{i=1}^{N} \beta^t r_i[t]$ with $\beta$ being the discount factor.

A control policy is effectively a function that takes the vector $(s_1[t], s_2[t], \ldots, s_N[t])$ as the input and produces the vector $(a_1[t], a_2[t], \ldots, a_N[t])$ as the output. It should be noted that the space of input is exponential in $N$. If each arm can be in one of $K$ possible states, then the number of possible inputs is $K^N$. This feature, which is usually referred to as the curse of dimensionality, makes finding the optimal control policy intractable.

### 3.2 THE WHITTLE INDEX

An index policy seeks to address the curse of dimensionality through decomposition. In each round, it calculates an index, denoted by $W_i(s_i[t])$, for each arm $i$ based on its current state. The index policy then selects the $M$ arms with the highest indices to activate. It should be noted that the index of an arm $i$ is independent from the states of any other arms.

Obviously, the performance of an index policy depends on the design of the index function $W_i(\cdot)$. A popular index with solid theoretical foundation is the Whittle index, which is defined below. Since we only consider one arm at a time, we drop the subscript $i$ for the rest of the paper.

Consider a system with only one arm, and a control policy that determines whether to activate the arm in each round $t$. Suppose that the policy needs to pay an activation cost of $\lambda$ every time it chooses to activate the arm. The goal of the control policy is to maximize the total discounted net reward, $\sum_{t=1}^{\infty} \beta^t (r[t] - \lambda a[t])$. The optimal control policy can be expressed by the set of states in which it would activate this arm for a particular $\lambda$, and we denote this set by $\mathcal{A}(\lambda)$. Intuitively, the higher the cost, the less likely the optimal control policy would activate the arm in a given state, and hence the set $\mathcal{A}(\lambda)$ should decrease monotonically. When an arm satisfies this intuition, we say that the arm is *indexable*.

**Definition 1** (Indexability). *An arm is said to be indexable if $\mathcal{A}(\lambda)$ decreases monotonically from the set of all states to the empty set as $\lambda$ increases from $-\infty$ to $\infty$. A restless bandit problem is said to be indexable if all arms are indexable.*

**Definition 2** (The Whittle Index). *If an arm is indexable, then its Whittle index of each state $s$ is defined as $W(s) := \sup_\lambda \{\lambda : s \in \mathcal{A}(\lambda)\}$.*

Even when an arm is indexable, finding its Whittle index can still be intractable, especially when the transition kernel of the arm is convoluted[1]. Our NeurWIN finds the Whittle index by leveraging the following property of the Whittle index: Consider the single-armed bandit problem. Suppose the initial state of an indexable arm is $s$ at round one. Consider two possibilities: The first is that the control policy activates the arm at round one, and then uses the optimal policy starting from round two; and the second is that the control policy does not activate the arm at round one, and then uses the optimal policy starting from round two. Let $Q_{\lambda,act}(s)$ and $Q_{\lambda,pass}(s)$ be the expected discounted net reward for these two possibilities, respectively, and let $D_s(\lambda) := \big(Q_{\lambda,act}(s) - Q_{\lambda,pass}(s)\big)$ be their difference. Clearly, the optimal policy should activate an arm under state $s$ and activation cost $\lambda$ if $D_s(\lambda) \geq 0$. We then have the following:

**Theorem 1.** *(Zhao, 2019, Thm 3.14) If an arm is indexable, then, for every state $s$, $D_s(\lambda) \geq 0$ if and only if $\lambda \leq W(s)$.*

---

[1]Niño-Mora (2007) described a generic approach for finding the Whittle index. The complexity of this approach is at least exponential to the number of states.

Our NeurWIN uses Thm. 1 to train neural networks that predict the Whittle index for any indexable arms. From Def. 1, a sufficient condition for indexability is when $D_s(\lambda)$ is a decreasing function. Thus, we define the concept of *strong indexability* as follows:

**Definition 3** (Strong Indexability). *An arm is said to be strongly indexable if $D_s(\lambda)$ is strictly decreasing in $\lambda$ for every state $s$.*

### 3.3 PROBLEM STATEMENT

We now formally describe the objective of this paper. We assume that we are given a simulator of one single restless arm as a black box. The simulator provides two functionalities: First, it allows us to set the initial state of the arm to any arbitrary state $s$. Second, in each round $t$, the simulator takes $a[t]$, the indicator function that the arm is activated, as the input and produces the next state $s[t+1]$ and the reward $r[t]$ as the outputs.

Our goal is to derive low-complexity index algorithms for restless bandit problems by training a neural network that approximates the Whittle index of each restless arm using its simulator. A neural network takes the state $s$ as the input and produces a real number $f_\theta(s)$ as the output, where $\theta$ is the vector containing all weights and biases of the neural network. Recall that $W(s)$ is the Whittle index of the arm. We aim to find appropriate $\theta$ that makes $|f_\theta(s) - W(s)|$ small for all $s$. Such a neural network is said to be *Whittle-accurate*.

**Definition 4** (Whittle-accurate). *A neural network with parameters $\theta$ is said to be $\gamma$-Whittle-accurate if $|f_\theta(s) - W(s)| \leq \gamma$, for all $s$.*

## 4 NEURWIN ALGORITHM: NEURAL WHITTLE INDEX NETWORK

In this section, we present NeurWIN, a deep-RL algorithms that trains neural networks to predict the Whittle indices. Since the Whittle index of an arm is independent from other arms, NeurWIN trains one neural network for each arm independently. In this section, we discuss how NeurWIN trains the Whittle index for one single arm.

### 4.1 CONDITIONS FOR WHITTLE-ACCURATE

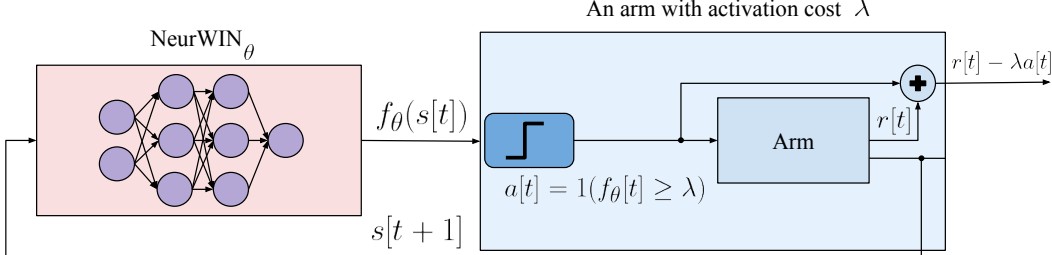

Figure 1: An illustrative motivation of NeurWIN.

Before presenting NeurWIN, we first discuss the conditions for a neural network to be $\gamma$-Whittle-accurate.

Suppose we are given a simulator of an arm and a neural network with parameters $\theta$. We can then construct an environment of the arm along with an activation cost $\lambda$ as shown in Fig. 1. In each round $t$, the environment takes the real number $f_\theta(s[t])$ as the input. The input is first fed into a step function to produce $a[t] = 1(f_\theta(s[t]) \geq \lambda)$, where $1(\cdot)$ is the indicator function. Then, $a(t)$ is fed into the simulator of the arm to produce $r[t]$ and $s[t+1]$. Finally, the environment outputs the net reward $r[t] - \lambda a[t]$ and the next state $s[t+1]$. We call this environment $Env(\lambda)$. Thus, the neural network can be viewed as a controller for $Env(\lambda)$. The following corollary is a direct result from Thm. 1.

**Corollary 1.** *If $f_\theta(s) = W(s), \forall s$, then the neural network with parameters $\theta$ is the optimal controller for $Env(\lambda)$, for any $\lambda$ and initial state $s[1]$. Moreover, given $\lambda$ and $s[1]$, the optimal discounted net reward is $\max\{Q_{\lambda,act}(s[1]), Q_{\lambda,pass}(s[1])\}$.*

Corollary 1 can be viewed as a necessary condition for a neural network to be 0-Whittle-accurate. Below, we establish a sufficient condition for $\gamma$-Whittle-accuracy.

**Theorem 2.** *If the arm is strongly indexable, then for any $\gamma > 0$ and an arbitrarily small positive constant $\delta$, there exists a positive $\epsilon$ such that the following statement holds: If, for any states $s_0, s_1$ and any activation cost $\lambda \in [f_\theta(s_0) - \delta, f_\theta(s_0) + \delta]$, the discounted net reward of applying a neural network to $Env(\lambda)$ with initial state $s_1$ is at least $\max\{Q_{\lambda,act}(s_1), Q_{\lambda,pass}(s_1)\} - \epsilon$, then the neural network is $\gamma$-Whittle-accurate.*

*Proof.* For a given $\gamma$, let $\epsilon = \min_s\{\min\{Q_{W(s)+\gamma,pass}(s) - Q_{W(s)+\gamma,act}(s), Q_{W(s)-\gamma,act}(s) - Q_{W(s)-\gamma,pass}(s)\}\}/2$. Since the arm is strongly indexable and $W(s)$ is its Whittle index, we have $\epsilon > 0$.

We prove the theorem by establishing the following equivalent statement: If the neural network is not $\gamma$-Whittle-accurate, then there exists states $s_0, s_1$, activation cost $\lambda \in [f_\theta(s_0) - \delta, f_\theta(s_0) + \delta]$, such that the discounted net reward of applying a neural network to $Env(\lambda)$ with initial state $s_1$ is strictly less than $\max\{Q_{\lambda,act}(s_1), Q_{\lambda,pass}(s_1)\} - \epsilon$.

Suppose the neural network is not $\gamma$-Whittle-accurate, then there exists a state $s'$ such that $|f_\theta(s') - W(s')| > \gamma$. We set $s_0 = s_1 = s'$. For the case $f_\theta(s') > W(s') + \gamma$, we set $\lambda = f_\theta(s') + \delta$. Since $\lambda > W(s') + \gamma$, we have $\max\{Q_{\lambda,act}(s'), Q_{\lambda,pass}(s')\} = Q_{\lambda,pass}(s')$ and $Q_{\lambda,pass}(s') - Q_{\lambda,act}(s') \geq 2\epsilon$. On the other hand, since $f_\theta(s') > \lambda$, the neural network would activate the arm in the first round and its discounted reward is at most

$$Q_{\lambda,act}(s') < Q_{\lambda,pass}(s') - 2\epsilon < \max\{Q_{\lambda,act}(s'), Q_{\lambda,pass}(s')\} - \epsilon.$$

For the case $f_\theta(s') < W(s') - \gamma$, a similar argument shows that the discounted reward for the neural network when $\lambda = f_\theta(s') - \delta$ is smaller than $\max\{Q_{\lambda,act}(s'), Q_{\lambda,pass}(s')\} - \epsilon$. This completes the proof. □

## 4.2 TRAINING PROCEDURES FOR NEURWIN

Thm. 2 states that a neural network that yields near-optimal net reward for any environments $Env(\lambda)$ is also Whittle-accurate. This observation motivates the usage of deep reinforcement learning to find Whittle-accurate neural networks. To make the output of the environments differentiable with respect to the input $f_\theta(s[t])$, we replace the step function in Fig. 1 with a sigmoid function $\sigma_m(f_\theta(s[t]) - \lambda) := (1 + exp(-m(f_\theta(s[t]) - \lambda)))^{-1}$, where $m$ is a sensitivity parameter. The environment then chooses $a[t] = 1$ with probability $\sigma_m(f_\theta(s[t]) - \lambda)$, and $a[t] = 0$ with probability $1 - \sigma_m(f_\theta(s[t]) - \lambda)$. We call this differentiable environment $Env^*(\lambda)$.

Our training procedure consists of multiple mini-batches, where each mini-batch is composed of a fixed number of episodes. At the beginning of each mini-batch, we randomly select two states $s_0$ and $s_1$. Motivated by the condition in Thm. 2, we consider the environment $Env^*(f_\theta(s_0))$ with initial state $s_1$ and aim to improve the empirical discounted net reward of applying the neural network to such an environment.

Our approach is based on the REINFORCE algorithm (Williams, 1992). In each episode $e$, we set $\lambda = f_\theta(s_0)$ and initial state to be $s_1$. We then apply the neural network with parameters $\theta$ to $Env^*(\lambda)$ and observe the sequences of actions $(a[1], a[2], \dots)$ and states $(s[1], s[2], \dots)$. We can use these sequences to calculate their gradients with respect to $\theta$ through backward propagation, which we denote by $h_e$. We also observe the discounted net reward and denote it by $G_e$. After all episodes in the mini-batch finish, we calculate the average of all $G_e$ as a bootstrapped baseline and denote it by $\bar{G}_b$. Finally, we do a weighted gradient ascent with the weight for episode $e$ being its offset net reward, $G_e - \bar{G}_b$. When the step size is chosen appropriately, the neural network will be more likely to follow the sequences of actions of episodes with larger $G_e$ after the weighted gradient ascent, and thus will have a better empirical discounted net reward. The complete algorithm is described in Alg. 1.

Obviously, the choice of $s_0$ and $s_1$ can have significant impact on the convergence speed of Alg. 1. In our implementation, we choose $s_0$ uniformly at random in each mini-batch. The choice of $s_1$ depends on the bandit problems. Some bandit problems naturally visit certain states far less frequently than other states. For such problems, we choose $s_1$ to be those less-frequently-visited states with higher probabilities, so as to ensure that Alg. 1 is able to learn the optimal control for these states. For other problems, we simply choose $s_1 = s_0$.

---

**Algorithm 1:** NeurWIN Training

---

**Input:** Parameters $\theta$, discount factor $\beta \in (0, 1)$, learning rate $L$, sigmoid parameter $m$
**Output:** Trained neural network parameters $\theta^+$
**for** *each mini-batch $b$* **do**
    Randomly choose $s_0$ and $s_1$, and set $\lambda \leftarrow f_\theta(s_0)$ ;
    **for** *each episode $e$ in the mini-batch* **do**
        Set the arm to state $s_1$, and set $h_e \leftarrow 0$ ;
        **for** *each round $t$ in the episode* **do**
            Choose $a[t] = 1$ w.p. $\sigma_m(f_\theta(s[t]) - \lambda)$, and $a[t] = 0$ w.p. $1 - \sigma_m(f_\theta(s[t]) - \lambda)$;
            **if** $a[t] = 1$ **then**
                $h_e \leftarrow h_e + \nabla_\theta \ln(\sigma_m(f_\theta(s[t]) - \lambda))$ ;
            **else**
                $h_e \leftarrow h_e + \nabla_\theta \ln(1 - \sigma_m(f_\theta(s[t]) - \lambda))$ ;
            **end**
        **end**
        $G_e \leftarrow$ empirical discounted net reward in episode $e$;
    **end**
    $L_b \leftarrow$ learning rate in mini-batch $b$;
    $\bar{G}_b \leftarrow$ the average of $G_e$ for all episodes in the mini-batch;
    Update parameters through gradient ascent $\theta \leftarrow \theta + L_b \sum_e (G_e - \bar{G}_b) h_e$ ;
**end**

---

## 5 EXPERIMENTS

### 5.1 OVERVIEW

In this section, we demonstrate NeurWIN's utility by evaluating it under three recently studied applications of restless bandit problems. In each application, we consider that there are $N$ arms and a controller can play $M$ of them in each round. We evaluate three different pairs of $(N, M)$: $(4, 1), (100, 10)$, and $(100, 25)$, and average the results of 200 independent runs when the problems are stochastic. Some applications consider that different arms can have different behaviors. For such scenarios, we consider that there are multiple types of arms and train a separate NeurWIN for each type. During testing, the controller calculates the index of each arm based on the arm's state and schedules the $M$ arms with the highest indices.

The performance of NeurWIN is compared against the proposed policies in the respective recent studies. In addition, we also implement and evaluate the REINFORCE algorithm (Williams, 1992) and the QWIC algorithm (Fu et al., 2019). The REINFORCE algorithm aims to find the optimal control by viewing a restless bandit problem as a Markov decision problem. Under this view, the number of states is exponential in $N$ and the number of possible actions is $\binom{N}{M}$. Thus, we are only able to evaluate REINFORCE for the case $N = 4$ and $M = 1$. The QWIC algorithm aims to find the Whittle index through Q-learning. It is a tabular method and does not scale well as the state space increases. Thus, we only evaluate QWIC when the size of the state space is small.

We use the same neural network architecture for NeurWIN in all three applications. The neural network is a fully connected one that consists of one input layer, one output layer, and two hidden layers. There are 16 and 32 neurons in the two hidden layers. The output layer has one neuron, and the input layer size is the same as the dimension of the state of one single arm. As for the REINFORCE algorithm, we choose the neural network architecture so that the total number of parameters is slightly more than $N$ times as the number of parameters in NeurWIN to make a fair

comparison. ReLU activation function is used for the two hidden layers. An initial learning rate $L = 0.001$ is set for all cases, with the Adam optimizer (Kingma & Ba, 2015) employed for the gradient ascent step. The discount factor is $\beta = 0.999$ and each mini-batch consists of five episodes.

For all cases, we implement the NeurWIN algorithm using PyTorch (Paszke et al., 2019), and train the agent on a single arm modelled after OpenAI's Gym API (Brockman et al., 2016). We provide a brief overview of each application and the experiment setting in the following sections. We refer readers to the appendices for detailed discussions on experiment settings.

## 5.2 RECOVERING BANDITS

The recovering bandits (Pike-Burke & Grunewalder, 2019) aim to model the time-varying behaviors of consumers. In particular, it considers that a consumer who has just bought a certain product, say, a television, would be much less interested in advertisements of the same product in the near future. However, the consumer's interest in these advertisements may recover over time. Thus, the recovering bandit models the reward of playing an arm, i.e., displaying an advertisement, by a function $f(\min\{z, z_{max}\})$, where $z$ is the time since the arm was last played and $z_{max}$ is a constant specified by the arm. There is no known Whittle index or optimal control policy for this problem.

The recent study (Pike-Burke & Grunewalder, 2019) on recovering bandit focuses on learning the function $f(\cdot)$ for each arm. Once it obtains an estimate of $f(\cdot)$, it uses a heuristic called $d$-lookahead to determine which arms to play. The $d$-lookahead policy enumerates all possible actions in the next $d$ rounds, and then pick the sequence of actions that yield that highest reward. Since the controller can choose $M$ arms out of $N$ arms to activate, with $\binom{N}{M}$ different possibilities, in each round, the complexity of the heuristic is $O\left(\binom{N}{M}^d\right)$ when $d > 1$. Thus, we are only able to evaluate 1-lookahead when $N = 100$. When $N = 4$ and $M = 1$, we evaluate 1-lookahead and 3-lookahead.

In our experiment, we consider that there are four types of arms and there are $\frac{N}{4}$ arms for each type. Different types of arms have different functions $f(\cdot)$. The state of each arm is its value of $\min\{z, z_{max}\}$ and we set $z_{max} = 20$ for all arms.

Experiment results are shown in Fig. 2. It can be observed that NeurWIN is able to outperform 1-lookahead in all settings with just a few thousands of training episodes. In contrast, for the case $N = 4$ and $M = 1$, REINFORCE only sees slight performance improvement over 50,000 training episodes and remains far worse than NeurWIN. This may be due to the explosion of state space. Even though $N$ is only 4, the total number of possible states is $20^4 = 160,000$, making it difficult for REINFORCE to learn the optimal control in just $50,000$ episodes. In contrast, since NeurWIN learns the Whittle index of each arm separately, its size of state space is only 20. QWIC performs poorly. This suggests that it does not learn a good approximation to the Whittle index.

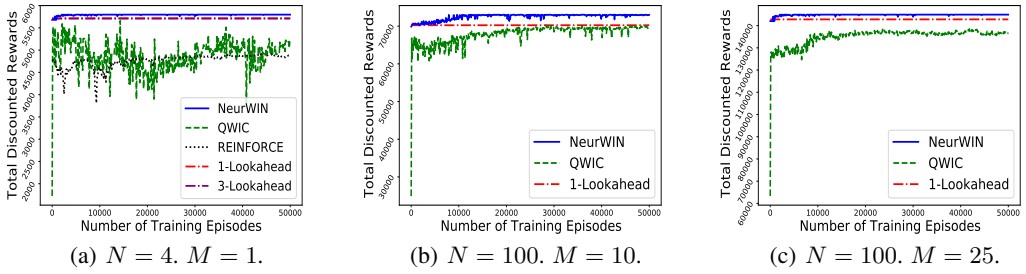

|     |     |     |
| --- | --- | --- |
| (a) $N = 4$. $M = 1$. | (b) $N = 100$. $M = 10$. | (c) $N = 100$. $M = 25$. |

Figure 2: Experiment results for the recovering bandits.

## 5.3 WIRELESS SCHEDULING

A recent paper (Aalto et al., 2015) studies the problem of wireless scheduling over fading channels. In this problem, each arm corresponds to a wireless client. Each wireless client has some data to be transmitted and it suffers from a holding cost of 1 unit per round until it has finished transmitting all its data. The channel quality of a wireless client, which determines the amount of data can be

transmitted if the wireless client is scheduled, changes over time. The goal is to minimize the sum of holding costs of all wireless clients. Equivalently, we view the reward of the system as the negative of the total holding cost.

Finding the Whittle index through theoretical analysis is difficult. Even for the simplified case when the channel quality is i.i.d. over time and can only be in one of two possible states, the recent paper (Aalto et al., 2015) can only derive the Whittle index under some approximations. It then proposes a *size-aware index* policy using its approximated index.

In the experiment, we adopt the settings of channel qualities of the recent paper. The channel of a wireless client can be in either a good state or a bad state. The amount of data that can be transmitted in a round is 33.6kb in a good state, and 8.4kb in a bad state. Initially, the amount of load is uniformly between 0 and 1Mb. The state of each arm is its channel state and the amount of remaining load. The size of state space is $2 \times 10^6$ for each arm. We consider that there are two types of arms, and different types of arms have different probabilities of being in the good state. We train a NeurWIN for each type. During testing, there are $\frac{N}{2}$ arms of each type.

Experiment results are shown in Fig. 3. It can be observed that NeurWIN is able to outperform the size-aware index policy with about $100,000$ training episodes. This result is significant when one considers the fact that the size-aware index is itself an approximation to the Whittle index. The experiment results thus suggest that NeurWIN is able to find a more accurate approximation to the Whittle index than the best known theoretical result. It can also be observed that REINFORCE performs poorly.

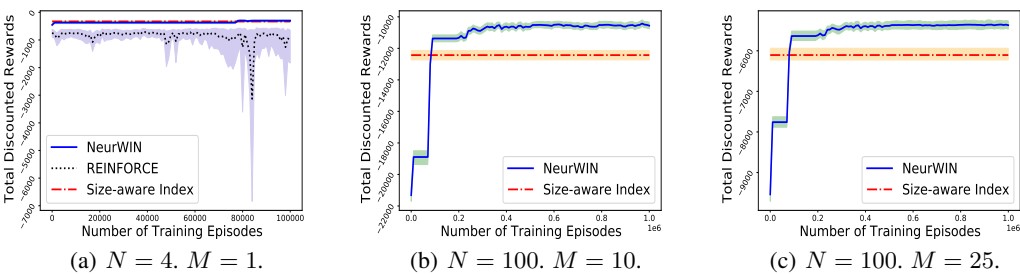

(a) $N = 4$. $M = 1$.    (b) $N = 100$. $M = 10$.    (c) $N = 100$. $M = 25$.

Figure 3: Average rewards and confidence bounds of different policies for wireless scheduling.

## 5.4 DEADLINE SCHEDULING

A recent study (Yu et al., 2018) proposes a deadline scheduling problem for the scheduling of electrical vehicle charging stations. In this problem, a charging station has $N$ charging spots and enough power to charge $M$ vehicles in each round. When a charging spot is available, a new vehicle may join the system and occupy the spot. Upon occupying the spot, the vehicle announces the time that it will leave the station and the amount of electricity that it needs to be charged. The charging station obtains a reward for each unit of electricity that it provides to a vehicle. However, if the station cannot fully charge the vehicle by the time it leaves, then the station needs to pay a penalty. The goal of the station is to maximize its net reward, defined as the difference between the amount of reward and the amount of penalty. Under an i.i.d. arrival assumption, the recent study has derived the precise characterization of the Whittle index, which we refer to as the deadline Whittle index. We further prove that this problem is strongly indexable in the appendix.

We use exactly the same setting as in the recent study (Yu et al., 2018) for our experiment. In this problem, the state of an arm is denoted by a pair of integers $(D, B)$, where $B$ is the amount of electricity that the vehicle still needs and $D$ is the time until the vehicle leaves the station. When a charging spot is available, its state is $(0, 0)$. $B$ is upper-bounded by 9 and $D$ is upper-bounded by 12. Hence, the size of state space is 109 for each arm.

The experiment results are shown in Fig. 4. It can be observed that the performance of NeurWIN converges to that of the deadline Whittle index in less than 500 training episodes.

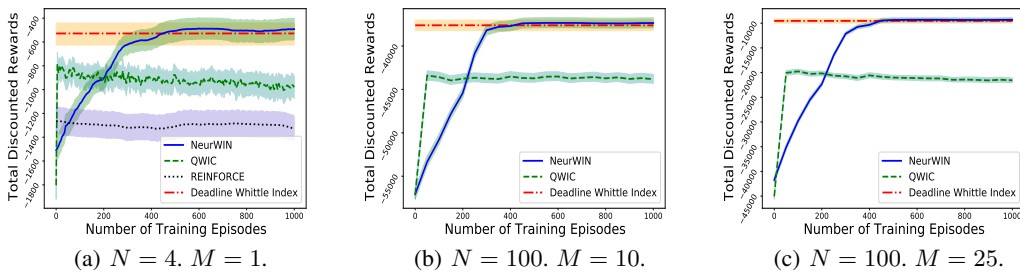

(a) $N = 4. M = 1.$       (b) $N = 100. M = 10.$       (c) $N = 100. M = 25.$

Figure 4: Average rewards and confidence bounds of different policies for deadline scheduling.

## 5.5 EXPERIMENT RESULTS WITH NOISY SIMULATORS

The training of NeurWIN requires a simulator for each arm. In this section, we evaluate the performance of NeurWIN when the simulator is not perfectly precise. In particular, let $R_{act}(s)$ and $R_{pass}(s)$ be the rewards of an arm in state $s$ when it is activated and not activated, respectively. Then, the simulator estimates that the rewards are $R'_{act}(s) = (1 + G_{act,s})R_{act}(s)$ and $R'_{pass}(s) = (1 + G_{pass,s})R_{i,pass}(s)$, respectively, where $G_{act,s}$ and $G_{pass,s}$ are independent Gaussian random variables with mean 0 and variance 0.05. In other words, the simulator has an average 5% error in its reward estimation.

We train NeurWIN using the noisy simulators for the recovering bandits problem and the deadline scheduling problem. For each problem, we compare the performance of NeurWIN against the respective baseline policies. Unlike NeurWIN, the baseline policies make decisions based on the true reward functions rather than the estimated ones. The results for the case $N = 100$ and $M = 25$ are shown in Fig. 5. It can be observed that NeurWIN is still able to achieve superior performance.

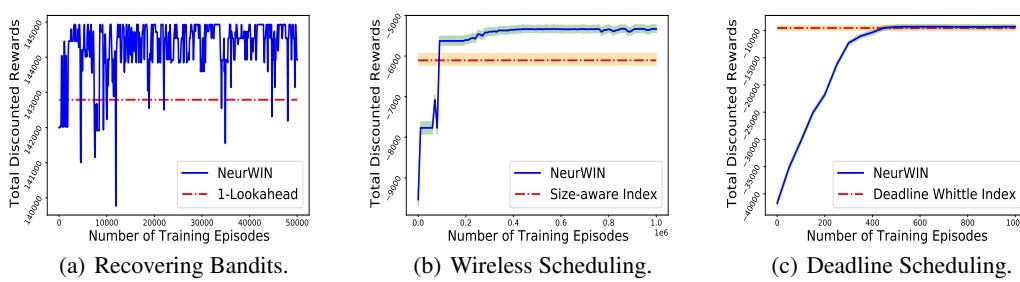

(a) Recovering Bandits.       (b) Wireless Scheduling.       (c) Deadline Scheduling.

Figure 5: Experiment results for NeurWIN with noisy simulators.

## 6 CONCLUSION

This paper introduced NeurWIN: a deep RL method for estimating the Whittle index for restless bandit problems. The performance of NeurWIN is evaluated by three different restless bandit problems. In each of them, NeurWIN significantly outperforms state-of-the-art control policies in terms of the total discounted reward.

NeurWIN can have important implications for restless bandit problems. There are many problems where the environments are well-defined, but the optimal control is not known. NeurWIN can obviously be used for such problems. For problems where the environments are not known a priori, NeurWIN nicely compliments existing studies that aim to learn the environments through online learning but fail to find the optimal control policy.

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

# A    RECOVERING BANDITS' TRAINING AND INFERENCE DETAILS

## A.1    FORMULATED RESTLESS BANDIT FOR THE RECOVERING BANDITS' CASE

We list here the terms that describes one restless arm in the recovering bandits' case:

**State** $s[t]$**:** The state is a single value $s[t] = z[t]$ called the waiting time. The waiting time $z[t]$ indicates the time since the arm was last played. The arm state space is determined by the maximum allowed waiting time $z_{max}$, giving a state space $\mathcal{S} := [1, z_{max}]$.

**Action** $a[t]$**:** As with all other considered cases, the agent can either activate the arm $a[t] = 1$, or not select it $a[t] = 0$. The action space is then $\mathcal{A} := \{0, 1\}$.

**Reward** $r[t]$**:** The reward is provided by the recovering function $f(z[t])$, where $z[t]$ is the time since the arm was last played at time $t$. If the arm is activated, the function value at $z[t]$ is the earned reward. A reward of zero if given if the arm is left passive $a[t] = 0$. Figure 6 shows the four recovering functions used in this work. The recovering functions are generated from,

$$f(z[t]) = \theta_0(1 - e^{-\theta_1 \cdot z[t]}) \tag{1}$$

Where the $\Theta = [\theta_0, \theta_1]$ values specify the recovering function. The $\Theta$ values for each class are given in table 1.

Table 1: $\Theta$ values used in the recovering bandits' case

| Class | $\theta_0$ **Value** | $\theta_1$ **Value** |
|-------|------------|------------|
| A | 10 | 0.2 |
| B | 8.5 | 0.4 |
| C | 7 | 0.6 |
| D | 5.5 | 0.8 |

**Next state** $s[t + 1]$**:** The state evolves based on the selected action. If $a[t] = 1$, the state is reset to $s[t + 1] = 1$, meaning that bandit's reward decayed to the initial waiting time $z[t + 1] = 1$. If the arm is left passive $a[t] = 0$, the next state becomes $s[t + 1] = \min\{z[t] + 1, z_{max}\}$.

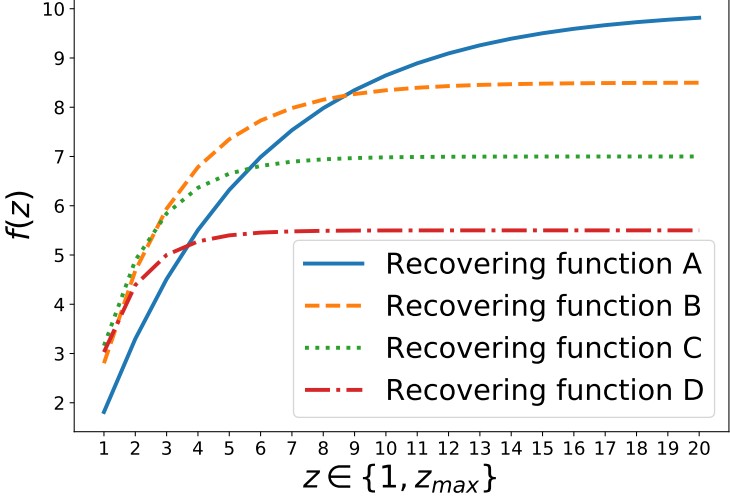

Figure 6: The selected recovering functions for the recovering bandits' case. For testing, we set each quarter of the instantiated $N$ arms to one of the shown $f(z)$ functions.

## A.2 TRAINING SETTING

The general training procedure for the NeurWIN algorithm is outlined in its pseudo code in section 4. Here we discuss the parameter selection and details specific to the recovering bandits' case. We train the neural network using NeurWIN for $50,000$ episode, and save the trained parameters at an episode interval of 100 episodes. The purpose of saving the parameters is to infer their control policies, and compare it with the 1-lookahead policy. In total, for $50,000$ training episodes, we end up with 500 models for inference. The selected neural network has 609 trainable parameters given as $\{1, 16, 32, 1\}$ layer neurons.

For training parameters, we select the sigmoid value $m = 5$, the episode's time horizon $T = 100$ timesteps, the mini-batch size to 5 episodes, and the discount factor $\beta = 0.999$. As with all other cases, each mini-batch of episodes has the same initial state $s[t = 1]$ which is provided by the arm. To ensure the agent experiences as many states in $[1, z_{max}]$ as possible, we set an initial state sampling distribution given as $Pr\{s[t = 1] = z\} = \frac{2^z}{2^1+2^2+...+2^{z_{max}}}$. Hence, the probability of selecting the initial state to be $s[t = 1] = z_{max}$ is 0.5. This initialization distribution allows the agent to experience the recovery function's awards at higher $z$ values.

At the agent side, we set the activation cost $\lambda$ at the beginning of each mini-batch. $\lambda$ is chosen to be the estimate index value $f_\theta(s')$ of a randomly selected state in $s' \in [1, z_{max}]$. The training continues as described in NeurWIN's pseudo code: the agent receives the state, and selects an action $a[t]$. If the agent activates the arm $a[t] = 1$, it receives a reward equal to the recovery function's value at $z$, and subtracts $\lambda$ from it. Otherwise, the reward $r[t]$ is kept the same for $a[t] = 0$. We note that no noise was added with the clean simulator, and the agent discounts the original reward value $\beta^t r[t] = \beta^t f(z[t])$. The process continues for all timesteps in the episode up to $T = 100$, and for remaining mini-batch episodes. A gradient ascent step is taken on the bootstrapped mini-batch return as described in section 4.

## A.3 INFERENCE SETTING

The inference setup measures NeurWIN's control policy for several $\binom{N}{M}$ settings. We test, for a single run, the control policies of NeurWIN and 1-lookahead over a time horizon $T = 3000$ timesteps. We set $N$ arms such that a quarter have one recovering function class from table 1. For example, when $N = 100$, 25 arms would have recovering function A that generates their rewards.

At each timestep, the 1-lookahead policy ranks the recovering functions reward values, and selects the $M$ arms with the highest reward values for activation. The incurred discounted reward at time $t$ is the discounted sum of all activated arms' rewards. The total discounted reward is then the discounted rewards over time horizon $T = 3000$. For inferring NeurWIN's control policy, we record the total discounted reward for each of the 500 models. An example testing procedure is as follows: we instantiate $N$ arms each having a neural network trained to $10,000$ episodes. At each timestep $t$, the neural networks provide the estimated index $f_{i,\theta}(s_i[t])$ for $i = 1, 2, \ldots, N$. The control policy activates the $M$ arms with the highest index values. The incurred discounted reward at time $t$ is the discounted sum of all activated arm's rewards $\beta^t R[t] = \beta^t \sum_{j=1}^{M} f_j(z[t])$. The same process continues for all timesteps in the horizon $T = 3000$. We then load the model parameters trained on $10,100$ episodes, and repeat the aforementioned testing process using the same seed values.

## A.4 REINFORCE TRAINING AND INFERENCE SETTING ON RECOVERING BANDITS

The REINFORCE algorithm was applied only the $\binom{N}{M}$ case where $N = 4$, and $M = 1$. For training, REINFORCE had four arms each with one of the recovery functions detailed in table 1. The training parameters are: initial learning rate $L = 0.001$, mini-batch size is 5 episodes, and a training episode time horizon $T = 100$ timesteps. Training was done up to $50,000$ episodes, where the trained parameters were saved at an interval of 100 episodes. The selected neural network had 2504 trainable parameters. This neural network size is larger than $609 \times 4 = 2436$ parameters of four NeurWIN neural networks.

For testing, the same procedure is followed as in A.3. The trained REINFORCE models were loaded, and each tested on the same arms as NeurWIN and deadline Whittle index policies. The testing was

made for all 500 trained model (each being trained up to a different episode count). The final control policy result was plotted along with NeurWIN and the 1-lookahead policies for $\binom{4}{1}$ arms.

### A.5 QWIC Training and Inference Setting on Recovering Bandits

The Q-learning Whittle Index Controller (QWIC) pseudo-code is given in (Fu et al., 2019). QWIC was trained in an offline setting using fixed $N$ restless arms with $M$ activations (i.e. $\binom{4}{1}$ $\binom{100}{10}$ $\binom{100}{25}$). QWIC selects from a set of candidate threshold values $\lambda \in \Lambda$ as index for each state. The algorithm learns Q function $Q \in \mathbb{R}^{\Lambda \times \mathcal{S} \times \{0,1\}}$. The estimated index $\tilde{\lambda}[s]$ per state $s$ is determined during training as,

$$\tilde{\lambda}[s] = \arg\min_{\lambda \in \Lambda} |Q(\lambda, s, 1) - Q(\lambda, s, 0)| \tag{2}$$

Hence, the converged index values and control performance depends on the initial set of candidate values $\Lambda$. We select $\Lambda$ to be 100 values evenly spaced in the interval $[0, 10]$. We note the set selection was based on NeurWIN's learned index values, which provides an advantage to QWIC training.

The exploration-exploitation trade-off is steered by parameter $\epsilon$. $\epsilon$ is initialized to $\epsilon_{max} = 1$, and decays with factor $\alpha = 0.01$ to $\epsilon_{min} = 0.01$. $\epsilon$ is updated at each timestep during training until it settles at $\epsilon_{min}$.

Other training parameters were selected as: initial learning rate $L = 0.001$, training episode time horizon of $T = 100$ timesteps, discount factor $\beta = 0.999$, . Training was done up to $50,000$ episodes, where the Q-learned indices $\bar{\Lambda}$ were saved at an interval of 100 episodes.

For testing, we use the same testing setting as in NeurWIN and REINFORCE. The learned indices are loaded for each training interval. In total, 500 estimated index mappings were tested for 200 independent runs, each trained up to a certain episode limit.

## B Wireless Scheduling Training and Inference Details

### B.1 Restless Arm Definition for the Wireless Scheduling Case

As with the recovering bandits' case, we first list the state $s[t]$, action $a[t]$, reward $r[t]$, and next state $s[t+1]$ that forms one restless arm:

**State $s[t]$:** The state is a vector $(y[t], v[t])$, where $y[t]$ is the arm's remaining load in bits, and $v[t]$ is the wireless channel's state indicator. $v[t] = 1$ means a good channel state and a higher transmission rate $r_2$, while $v[t] = 0$ is a bad channel state with a lower transmission rate $r_1$.

**Action $a[t]$:** The agent either activates the arm $a[t] = 1$, or keeps it passive $a[t] = 0$. The reward and next state depend on the chosen action.

**Reward $r[t]$:** The arm's reward is the negative of *holding cost* $\psi$, which is a cost incurred at each timestep for not completing the job. If the selected action $a[t] = 1$, then the reward at time $t$ is $r[t] = -\psi - \lambda$. Otherwise, reward is just $r[t] = -\psi$.

**Next state $s[t+1]$:** The next state evolves differently as given below,

$$s[t+1] = \begin{cases} (y[t] - r_2, 1) & \text{if } q(v[t]) = 1, a[t] = 1 \\ (y[t] - r_1, 0) & \text{if } q(v[t]) = 0, a[t] = 1 \\ (y[t], q(v[t])) & \text{otherwise} \end{cases} \tag{3}$$

Where $q(v[t])$ is the probability of a good channel state.

## B.2 TRAINING SETTING

We again emphasize that NeurWIN training happens only on one restless arm. The general training procedure was described in NeurWIN's pseudo code. This discussion pertains only to the wireless scheduling case.

The neural network has 625 trainable parameters given as $\{2, 16, 32, 1\}$ neuron layers. The training happens for $1,000,000$ episodes, and we save the model parameters at each 1000 episodes. Hence, the training results in 1000 models trained up to different episode limit.

For the wireless scheduling case, we set the sigmoid value $m = 0.01$, mini-batch size to 5 episodes, and the discount factor to $\beta = 0.999$. Episode time horizon is dependent on the remaining job size $y[t]$. The episode terminates either if $y[t] = 0$ or $t = 3000$. The holding cost is set to $c = 1$, which is incurred for each timestep the job is not completed. We also set the good transmission rate $r_2 = 33.6$ kb, and the bad channel transmission rate $r_1 = 8.4$ kb. During training, the good channel probability is $q(v[t]) = 0.5$.

The episode defines one job size sampled uniformly from the range $y[t = 1] \sim (0, 1 \text{ Mb}]$. All episodes in one mini-batch have the same initial state, as well as the same sequence of good channel states $[v[t = 1], v[t = 2], \ldots, v[t = T]]$.

At the agent side, NeurWIN receives the initial state $s[t = 1]$, and sets the activation cost $\lambda = f_\theta(s[t = 1])$ for all timesteps of all mini-batch episodes. As mentioned before, we save the trained model at an interval of 1000 episodes. For $1,000,000$ episodes, this results in 1000 models trained up to their respective episode limit.

## B.3 INFERENCE SETTING

For testing, the aim is to measure the trained models' control performance against the size-aware index. We instantiate $N$ arms and activate $M$ arms at each timestep $t$ until all users' jobs terminate. We average the total discounted reward for all control policies over 200 independent inference runs. Half of the arms have a good channel probability $q(v[t]) = 0.75$. The other half has a good channel probability $q(v[t]) = 0.1$.

We compare NeurWIN's control policy at different training episodes' limits with the size-aware index policy. The size-aware index is defined as follows: at each timestep, the policy prioritizes arms in the good channel state, and calculates their *secondary index*. The secondary index $\hat{v}_i$ of arm $i$ state $(y_i[t], v_i[t])$ is defined as,

$$\hat{v}_i(y_i[t], v_i[t]) = \frac{c_i r_{i,2}}{y_i[t]} \tag{4}$$

The size-aware policy then activates the highest $M$ indexed arms. In case the number of good channel arms is below $M$, the policy also calculate the *primary index* of all remaining arms. The primary index $v_i$ of arm $i$ state $(y_i[t], v_i[t])$ is defined as,

$$v_i(y_i[t], v_i[t]) = \frac{c_i}{q_i[t](r_{i,2}/r_{i,1}) - 1} \tag{5}$$

Rewards received from all arms are summed, and discounted using $\beta = 0.999$. The inference phase proceeds until all jobs have been completed.

For NeurWIN's control policy, we record the total discounted reward for the offline-trained models. For example, we set $N$ arms each coupled with a model trained on $10,000$ episodes. The models output their arms' indices, and the top $M$ indexed arms are activated. In case the remaining arms are less than $M$, we activate all remaining arms at timestep $t$. timestep reward $\beta^t R[t] = \beta^t \sum_{i=1}^{N} r[t]$ is the sum of all arms' rewards. Once testing for the current model is finished, we load the next model $11,000$ for each arm, and repeat the process. We note that the arms' initial loads are the same across runs, and that the sequence of good channel states is random.

## B.4 REINFORCE TRAINING AND INFERENCE SETTING ON WIRELESS SCHEDULING

The REINFORCE algorithm was applied only the $\binom{4}{1}$ case. The four arms have the same training setting as described in section B.2. The training parameters are: initial learning rate $L = 0.001$,

mini-batch size is 5 episodes, and good channel probability for all four arms $q(v[t]) = 0.5$. The episode time horizon has a hard limit of $\bar{T} = 3000$ timesteps. However, an episode can terminate if all arms' loads were fully processed (i.e. $\sum_{i=1}^{4} y_i[t] = 0$). Training was done up to $100,000$ episodes, where the trained parameters were saved at an interval of $1000$ episodes. The selected neural network had $2532$ trainable parameters so to have slightly more parameters than four NeurWIN neural networks.

For testing, the same procedure is followed as in B.3. The trained REINFORCE models were loaded, and each tested on the same arms as NeurWIN and size-aware index. The final control policy result was plotted along with NeurWIN and Whittle index policy for the $\binom{4}{1}$ testing setup.

## C  DEADLINE SCHEDULING TRAINING AND INFERENCE DETAILS

### C.1  FORMULATED RESTLESS BANDIT FOR THE DEADLINE SCHEDULING CASE

The state $s[t]$, action $a[t]$, reward $r[t]$, and next state $s[t+1]$ of one arm are listed below:

**State $s[t]$:** The state is a vector $(D, B)$. $B$ denotes the job size (i.e. amount of electricity needed for an electric vehicle), and $D$ is the job's time until the hard drop deadline $d$ is reached (i.e. time until an electric vehicle leaves).

**Action $a[t]$:** The agent can either activate the arm $a[t] = 1$, or leave it passive $a[t] = 0$. The next state changes based on two different transition kernels depending on the selected action. The reward is also dependent on the action at time $t$.

**Reward $r[t]$:** The agent, at time $t$, receives a reward $r[t]$ from the arm,

$$r[t] = \begin{cases} (1-c)a[t] & \text{if } B[t] > 0, D[t] > 1 \\ (1-c)a[t] - F(B[t] - a[t]) & \text{if } B[t] > 0, D[t] = 1 \\ 0 & \text{otherwise} \end{cases} \quad (6)$$

Where $c$ is a constant processing cost incurred when activating the arm, $F(B[t] - a[t])$ is the *penalty function* for failing to complete the job before $D = 1$. The penalty function was chosen to be $F(B[t] - a[t]) = 0.2(B[t] - a[t])^2$.

**Next state $s[t+1]$:** The next state $D[t+1]$ decreases by one, while the job size $B$ depends on the selected action as,

$$s[t+1] = \begin{cases} (D[t] - 1, B[t] - a[t]) & \text{if } D[t] > 1 \\ (D, B) \text{ with prob. } Q(D, B) & \text{if } D[t] \leq 1 \end{cases} \quad (7)$$

Where $Q(D, B)$ is the arrival probability of a new job (i.e. a new electric vehicle arriving at a charging station) if the position is empty. For training and inference, we set $Q(D, B) = 0.7$.

### C.2  STRONG INDEXABILITY PROOF FOR THE DEADLINE SCHEDULING CASE

It has been shown that the Whittle index for this problem is,

$$v(D, B) := \begin{cases} 0 & \text{if } B = 0 \\ 1 - c & \text{if } 1 \leq B \leq D - 1 \\ \begin{aligned} & \beta^{D-1}F(B - D + 1) \\ & -\beta^{D-1}F(B - D) + 1 - c \end{aligned} & \text{if } D \leq B \end{cases} \quad (8)$$

We further demonstrate that this problem is strongly indexable.

**Theorem 3.** *The restless bandit for the deadline scheduling problem is strongly indexable.*

*Proof.* Fix a state $s = (D, B)$, the function $D_s(\lambda) := (Q_{\lambda,act}(s) - Q_{\lambda,pass}(s))$ is a continuous and piece-wise linear function since the number of states is finite. Thus, it is sufficient to prove that $D_s(\lambda)$ is strictly decreasing at all points of $\lambda$ where $D_s(\lambda)$ is differentiable. Let $L_{\lambda,act}(s)$ be the sequence of actions taken by a policy that activates the arm at round 1, and then uses the optimal policy starting from round 2. Let $L_{\lambda,pass}(s)$ be the sequence of actions taken by a policy that does not activate the arm at round 1, and then uses the optimal policy starting from round 2. We prove this theorem by comparing $L_{\lambda,act}(s)$ and $L_{\lambda,pass}(s)$ on every sample path. We consider the following two scenarios:

In the first scenario, $L_{\lambda,act}(s)$ and $L_{\lambda,pass}(s)$ are the same starting from round 2. Let $b$ be the remaining job size when the current deadline expires under $L_{\lambda,act}(s)$. Since $L_{\lambda,pass}(s)$ is the same as $L_{\lambda,act}(s)$ starting from round 2, its remaining job size when the current deadline expires is $b + 1$. Thus, $D_s(\lambda) = 1 - c - \lambda + \beta^{D-1}(F(b+1) - F(b))$, which is strictly decreasing in $\lambda$ whenever $D_s(\lambda)$ is differentiable.

In the second scenario, $L_{\lambda,act}(s)$ and $L_{\lambda,pass}(s)$ are not the same after round 2. Let $\tau$ be the first time after round 2 that they are different. Since they are the same between round 2 and round $\tau$, the remaining job size under $L_{\lambda,act}(s)$ is no larger than that under $L_{\lambda,pass}(s)$. Moreover, the Whittle index is increasing in job size. Hence, we can conclude that, on round $\tau$, $L_{\lambda,pass}(s)$ activates the arm and $L_{\lambda,act}(s)$ does not activate the arm. After round $\tau$, $L_{\lambda,act}(s)$ and $L_{\lambda,pass}(s)$ are in the same state and will choose the same actions for all following rounds. Thus, the two sequences only see different rewards on round 1 and round $\tau$, and we have $D_s(\lambda) = (1 - c - \lambda)(1 - \beta^{\tau-1})$, which is strictly decreasing in $\lambda$ whenever $D_s(\lambda)$ is differentiable.

Combining the two scenarios, the proof is complete. $\qquad\square$

## C.3 TRAINING SETTING

NeurWIN training is made for 1000 episodes on the deadline scheduling case. We save the trained model parameters at an interval of 5 episodes for inferring the control policy after training. Hence, the training produces 200 different set of parameters that output the estimated index given their respective training limit. The neural network had 625 trainable parameters given as $\{2, 16, 32, 1\}$, where the input layer matches the state size.

For the deadline scheduling training, we set the sigmoid value $m = 1$, episode's time horizon $T = 3000$ timesteps, mini-batch size to 5 episodes, and the discount factor $\beta = 0.999$. The processing cost $c = 0.5$, with the job arrival rate $Q(D, B) = 0.7$. Training procedure follows section 4.2 from the main text. The arm randomly picks an initial state $s[t = 1] = (D, B)$, with a maximum $\bar{D} = 12$, and maximum $\bar{B} = 9$. The arm fixes the initial states across episodes in the same mini-batch for proper return comparison. The sequence of job arrivals in an episode's horizon is also fixed across a mini-batch. For example, one episode in mini-batch 1 would have the sequence $[(11, 5), (6, 2), (8, 4), \ldots, (3, 5)]$, then all other episodes in the same mini-batch would pass the same sequence. This way, the actions taken by the agent would be the critical factor in comparing a mini-batch return, and ultimately in tuning the estimated index value $f_\theta(\cdot)$.

At the agent side, NeurWIN receives the initial state $s[t = 1]$, sets the activation cost $\lambda = f_\theta(s[t = 1])$. This activation cost $\lambda$ selection method hence depends on the current network parameters $\theta$, which are modified after every gradient ascent step. Training follows as described in NeurWIN's pseudo code.

In figure 7, we plot the trained NeurWIN index for all possible state enumerations of $\bar{B} = 9$ and $D \in \{1, 2, 3\}$. The output index from the untrained neural network is also plotted for convergence comparison.

In figure 8, the trained restless bandit indices for noisy reward function is given. All possible states in $\bar{B} = 9$ for $D \in \{1, 2, 3\}$. For $\mathcal{N}(0, 0.05)$ added noise per timestep, the learned indices still match the state ordering found when trained with the true reward function.

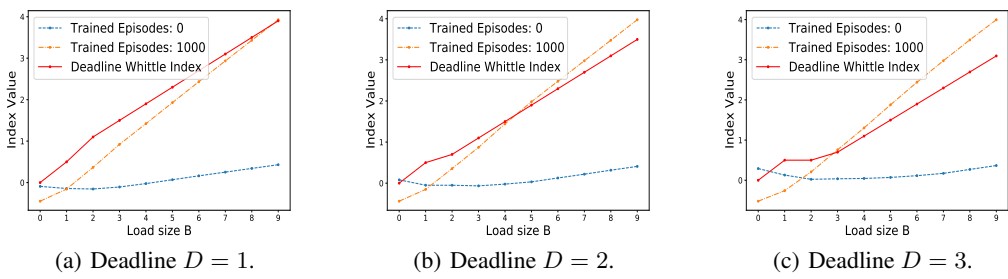

(a) Deadline $D = 1$.       (b) Deadline $D = 2$.      (c) Deadline $D = 3$.

Figure 7: Trained indices using the true reward function.

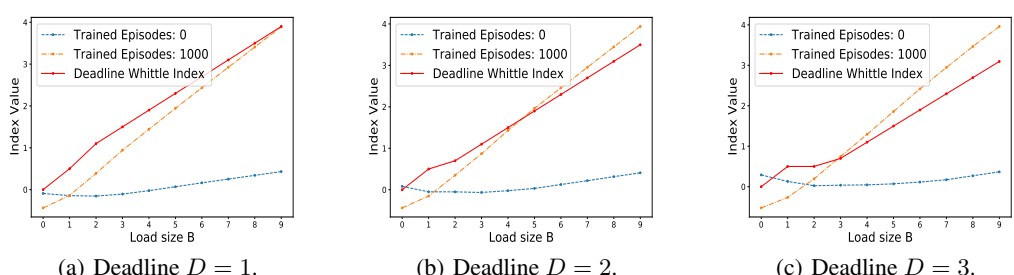

(a) Deadline $D = 1$.      (b) Deadline $D = 2$.      (c) Deadline $D = 3$.

Figure 8: Trained indices using the noisy reward function.

## C.4 INFERENCE SETTING

In order to infer the resultant control policy, we are required to test the performance on models saved at different episodes' intervals. In other words, the trained models' parameters are tested at an interval of episodes, and their discounted rewards are plotted for comparison.

From the trained models described in C.3, we instantiate $N$ arms, and activate $M$ arms at each timestep. The inference step compares the resultant control policy with the deadline Whittle index $v(D, B)$.

The testing is done for a time horizon of $T = 3000$ timesteps. The queue, modelled as $N$ restless arms, has $M$ positions activated at each timestep. Each arm has a unique sequence of job arrivals from other arms that differentiates its index value. For the deadline Whittle index, we calculate the indices according to 8, and activate the highest $M$ indices-associated arms. The accumulated reward from all arm (activated and passive) is then discounted with $\beta$.

For NeurWIN control policy, we instantiate $N$ arms, and test the trained models up to a given episode. For example, we load a NeurWIN model trained for 100 episodes on one arm, and set $N$ arms each with its own trained agent on 100 episodes. Once the testing is complete, we load the next model trained at 105 episodes, and repeat the process for 105 episodes. The final result is NeurWIN's control policy's performance on $N$ arms given the models' training.

We perform the testing over 200 independent runs up to 1000 episodes, where each run the arms are seeded differently. We stress that both the deadline Whittle index and NeurWIN policies were applied on identical seeded arms across the 200 runs. Meaning the sequence of arrivals and rewards experienced was fixed for each arm in each run. Results were provided in the main text for this setting.

## C.5 REINFORCE TRAINING AND INFERENCE SETTING ON DEADLINE SCHEDULING

The REINFORCE algorithm was applied on the $\binom{4}{1}$ testing case. For training, REINFORCE was trained on the same training setting as described in C.3 with the same parameters when appropriate.

The four restless arms were seeded differently to give unique job sequences. Training was made until 1000 episodes, where the trained parameters were saved at an interval of 5 episodes. The selected neural network had 2532 trainable parameters. The REINFORCE parameters' count are purposefully slightly larger than $625 \times 4 = 2500$ parameters of four NeurWIN neural networks.

For testing, the same procedure is followed as explained in C.4. The trained REINFORCE models were loaded, and each tested on the same arms as NeurWIN and deadline Whittle index policies. The testing was made for all 200 trained model (each being trained up to a different episode count). The final control policy result was plotted along with NeurWIN and Whittle index policy for $\binom{4}{1}$ arms.

### C.6    QWIC TRAINING AND INFERENCE SETTING ON DEADLINE SCHEDULING

QWIC was trained in an offline setting for the sets $\binom{4}{1}$ $\binom{100}{10}$ $\binom{100}{25}$. We select the same candidate set $\Lambda$ as in the recovering bandits case, which is 100 values evenly spaced in the interval $[0, 10]$. $\epsilon$ was initialized to $\epsilon_{max} = 1$, and decays with factor $\alpha = 0.01$ to $\epsilon_{min} = 0.01$. $\epsilon$ is updated at each timestep during training until it decays to $\epsilon_{min}$.

Other training parameters: initial learning rate $L = 0.001$, training episode time horizon of $T = 3000$ timesteps, discount factor $\beta = 0.999$, . Training was done up to $1,000$ episodes, where the select q-learned indices $\bar{\Lambda}$ were saved at an interval of 5 episodes. We test the Q-learning indices using the same setting as NeurWIN and REINFORCE. The estimated index mappings were tested for 200 independent runs.

We refer the reader to the code for further implementation details.

