# OpenReview forum: "NeurWIN: Neural Whittle Index Network for Restless Bandits via Deep RL"
_ICLR.cc/2021/Conference — Reject_

### Official Review · AnonReviewer4 · 2020-10-21
**The experiment on recovering bandits is not convincing**

**Rating:** 4
**Confidence:** 4

**Review:**

This paper proposes the use of Deep Learning, namely a multi-layer perceptron, for approximating the Whittle index in restless bandits.

The introduction and the related works are well written. The problem and the background on restless bandits are clearly exposed.

However, there are a lot of issues in the presentation of the proposed algorithm, NeurWin, in the analysis of the proposed algorithm, and in the experimentations.

The statement of Corollary 1 is not correct. In the general case, the Whittle index is not optimal, it is a heuristic. So if the neural controller produces the Whittle index, it does not produce the optimal discounted reward. As the authors suggest the statement of Corollary 1 should be rewritten as a necessary condition for a neural network to be Whittle-accurate.

The proof of Theorem 2 is wrong.
The statement of Theorem 2 is for any states s_0,s_1, for any \lambda \in [f_\theta(s_0)-\delta, f_\theta(s_0) + \delta ] … then the neural network is \gamma-Whittle-accurate.
In the proof the authors only show that the neural network is \gamma-Whittle-accurate, when s_0=s_1 and when \lambda =,f_\theta(s_0) + \delta. So they cannot conclude that Theorem 2 holds for any states s_0,s_1, and for any \lambda \in [f_\theta(s_0)-\delta, f_\theta(s_0) + \delta ].

In the section 4.2 where the training procedure is described the authors write that the choice of s_1 depends on the MAB problems, and hence it has be chosen knowing the MAB problem. The hearth of MAB problem is precisely that you do not know if some states or some arms have to be less-frequently-visited that others. This is the exploration / exploitation dilemma.
So if for tuning the proposed algorithm you need to know the MAB problem, the proposed algorithm does not work at all.

In the experimental section the experiment on recovering bandits is not convincing.
The authors write that the algorithmic complexity of the algorithms proposed in recovering bandits is (\binom(N,M))^d. It is not correct. In recovering bandits the arms are the recovering functions. In your experiment it should be N, and d is the number of times the arms are sampled in a round, so in your experiment it should be M. The algorithmic complexity is N^M, that it is still very large. However in the cited paper the authors propose the use of optimistic planning with a given budget B. So the algorithmic complexity of their algorithms is B, which is a parameter.


So the choice of d=1 in the experiment is unfair.
Moreover, in the experiment the choice of only four different recovering functions for 100 arms is a little bit strange. In recovering bandits the arms are the recovering functions.

Finally, the choice of training offline the neural network is not realistic in a bandit setting, where the environment is not known at the beginning. To make a fair comparison with other bandit algorithms the authors should train the neural network online and compare the accumulated rewards taking into account the convergence time to a good solution.

Approximating the Whittle index by a multi-layer perceptron is a good idea, but the submitted paper is not ready for publication.

__________________________________________________________

After rebuttal, the authors reformulate Theorem 2, and then I think it is right. I raised my score.
I am still not convinced by their experiments on recovering bandits.

---

> ### Author Response · Authors · 2020-11-12
> **Clarifications of misunderstandings.**
>
> It appears that the reviewer has some misunderstandings about the paper. We would like to clarify the issues and better explain them in the paper.
>
> 1. Regarding Corollary 1:
> When one considers that there is only one arm with an activation cost, then Whittle index IS the optimal control policy as long as the arm is indexable. This is the direct result from Theorem 1. It seems that the "general case" under which the reviewer claims that Whittle index policy may not be optimal is the case when there are multiple arms, and the controller chooses to activate the arms with the highest indices. In Section 4, we are focused on training the Whittle index for one single arm, as shown in Fig. 1. Hence, Theorem 1 and Corollary 1 hold. We will revise the paper to make it clear that we consider one single arm at the beginning of Section 4.
>
> 2. Regarding the proof of Theorem 2:
> We would like to explain the proof procedure. Theorem 2 states that "If the condition holds for any (s_0, s_1, \lambda), then the neural network is \gamma-Whittle-accurate." In the proof, we prove the equivalent statement: "If the neural network is NOT \gamma-Whittle-accurate, then there exists at least one (s_0, s_1, \lambda) that violates the condition." Indeed, the proof explains how to find the (s_0, s_1, \lambda) that violates the condition, and hence the theorem holds. We will revise the paper so that it is clear that we are proving the equivalent statement at the beginning of the proof.
>
> 3. Regarding the complexity of the recovering bandits:
> It seems the issue is that the reviewer misunderstands the d-lookahead algorithm since it is not clearly described in our paper. The variable d is NOT "the number of times the arms are sampled in a round", as the reviewer claims. It is the number of steps that the d-lookahead algorithm enumerates. Each step consists of choosing M arms out of N arms, with a total of (N \choose M) possible choices. Therefore, in d steps, there are a total of (N \choose M)^d different choices. In the recovering bandit paper, it considers the case with N = K and M = 1. In the beginning of Section 6 of the recovering bandit paper, it indeed states that "Algorithm 1...searches K^d leaves". Note that K^d = (K \choose 1)^d. This is consistent with our complexity analysis.
>
> We would also like to point out that the trick used in the recovering bandit paper to improve computational efficiency is not sufficient for our case. In Section 7, the recovering bandit paper states that the trick still requires to evaluate ~0.1% of all possible choices. For the case N = 100, M = 25, d = 2, there are more than 5*10^46 possible choices. Even evaluating only 0.1% of them is intractable. Thus, we are only able to present the result for d = 1. Please note that, when d =1, the d-lookahead algorithm still activates M arms in each step. Hence, the comparison is fair.
>
> Seeing the confusion arisen, we will update the paper to include a short description of the d-lookahead algorithm.
>
> 4. Regarding offline training:
> We would like to stress that this paper studies *restless* multi-armed bandit problem. For restless bandits, the optimal control policy is difficult to find even when the environment is known. Indeed, there are many problems where the environments are well-defined, but the optimal control is not known, such as the wireless scheduling problem described in Section 5.3. We believe that our work makes an important contribution to these problems. For example, we show in Section 5.3 that NeurWIN achieves better reward than the best-known policy. For problems where the environments are not known a priori, we believe that our work nicely compliments existing studies that aim to learn the environments but fail to find the optimal control policy. For example, the recovering bandit paper only uses the heuristic d-lookahead after learning the environment. Our work would compliment the recovering bandit paper by finding the near-optimal control policy after it learns the environment.
>
> We will add some discussions to highlight the usefulness of this work.

---

> > ### Comment · AnonReviewer4 · 2020-11-18
> > **Re: Clarifications of misunderstandings.**
> >
> > I thank the authors for answering my concerns.
> >
> > Concerning Corollary 1, ok.
> >
> > Concerning Theorem 2, the problem I raised comes from the statement that is misleading:
> >
> > As the sentence “for any activation cost \lambda \in [f_(s0) -\delta; f_(s0) + \delta]” is separated by a coma and positioned after the condition “at least Max (Q_\lambda,act(s1),Q_\lambda,pass(s1)) - \epsilon”, it  means that “for any activation cost \lambda \in [f_(s0) -\delta; f_(s0) + \delta]” is another condition and not the domain of \lambda.
> >
> > Given your answer I understood that it is not the case. I suggest you to modify the statement in Theorem 2.
> >
> > Concerning the recovering bandits, for claiming that "d is the number of times the arms are sampled in a round", I read Algorithm 1 in (Pike-Burke & Grunewalder, 2019). At each time step t, d arms are played, and the posterior distributions of the played arms are updated after the d arms are sampled.
> > An arm can be selected several times per time step (section 5.2 in the referred paper) or it can be selected only once (section 5.1 in the referred paper). I think this last case corresponds to your experiment: selecting M arms out of N at each time step.
> > If it is possible given the values of N and M in your experiments, I suggest you to test recovering bandits in this way.

---

> > > ### Author Response · Authors · 2020-11-22
> > > **Responses**
> > >
> > > Regarding Theorem 2: We thank the reviewer for the feedback. We revised the statement for better clarity. It now reads:
> > >
> > > If, for any states $s_0,s_1$ and any activation cost $\lambda\in[f_\theta(s_0)-\delta, f_\theta(s_0)+\delta]$, the discounted net reward of applying a neural network to $Env(\lambda)$ with initial state $s_1$ is at least $\max\{ Q_{\lambda, act}(s_1), Q_{\lambda, pass}(s_1)\}-\epsilon$, then the neural network is $\gamma$-Whittle-accurate.
> > >
> > > Regarding recovering bandit:
> > >
> > > We think the reviewer may be referring to the line "for l = 0,...,d-1 do" in Alg. 1 in the recovering bandit paper. We want to emphasize that this for loop activates d arms in d time steps, and therefore only activates 1 arm in each time step. This can be observed by noting the following:
> > >
> > > 1. In the line below the for loop, the reward is Y_{J_l,t+l}. The subscript is t+l instead of t because the l-th arm is played at time step t+l.
> > >
> > > 2. In the initialization step, the algorithm states that it only makes decisions at times t = 1, d+1, 2d+1, .... Thus, for each time t, the algorithm needs to determine which arm to activate in each of the next d time steps. This is why the algorithm ends up choosing d arms.
> > >
> > > In summary, d is a parameter for the algorithm, not a parameter for the problem. No matter what the value of d is, the problem only allows one arm to be activated in each time step.
> > >
> > > In contrast, our problem allows the controller to activate M arms in each time step. Hence, the recovering bandit paper is the special case when M = 1. We expanded Alg. 1 in the recovering bandit paper to handle the case M > 1. Results reported in our paper is based on the expanded 1-lookahead algorithm that activates M arms in each time step. We would like to emphasize that M and d are two different and unrelated parameters.

---

> > > > ### Comment · AnonReviewer4 · 2020-11-24
> > > > **in red is indeed a parameter of the algorithm but only because choosing the**
> > > >
> > > > I read your answer. Ok for the new statement of Theorem 2. To be fair, I will raise my score.
> > > >
> > > > Concerning recovering bandits, I do not agree.
> > > >
> > > > There are two points in recovering bandits:
> > > >
> > > > 1/ The rewards of the arm depend on the time since the arm was last played, which is modeled by a recovering function.
> > > >
> > > > 2/ Because of this, the optimal policy is the deterministic policy which knows the recovery functions and T, and using this selects the best sequence of T arms.
> > > >
> > > > However, the optimal policy is untractable. That is why in recovering bandits, the author focus on minimizing the d-step Lookahead regret. So d is not a parameter of the algorithm, but is problem dependent.
> > > >
> > > > In Algorithm 1, the important point is: when are the posterior distributions of the played arms  updated with the observed rewards ?
> > > >
> > > > After playing d arms.
> > > >
> > > > In the loop l= 0, . . . , d − 1, the rewards are not used to choose the next arm to play. This corresponds to play d-arms and then to observe the rewards.
> > > > So the reviewer does not understand why you choose d=1, while you could choose an higher value for d.

---

> > > > > ### Author Response · Authors · 2020-11-25
> > > > > **Regarding recovering bandits**
> > > > >
> > > > > Regarding recovering bandits:
> > > > >
> > > > > We think there may be some mutual misunderstandings concerning the term "d-lookahead". In the recovering bandits, there are "d-step lookahead regret", "d-step lookahead UCB", "d-step lookahead oracle", etc. To be clear, the "d-lookahead" in our paper is the "d-step lookahead oracle", which is also called "optimal d-step lookahead policy", in the recovering bandit paper.
> > > > >
> > > > > Now, we have the following responses to the reviewer's concern:
> > > > >
> > > > > 1. As shown in Fig. 1(b) in the recovering bandit paper, the d-step lookahead policy needs to first build a d-step lookahead tree and evaluates the rewards in each of the leaf nodes. In the context of the recovering bandit paper, the number of leave nodes is K^d (also shown in Fig. 1(b)). In the context of our paper, the number of leave nodes is (N\choose M)^d.
> > > > >
> > > > > 2. Thus, we are unable to do d>1 when (N,M) = (100,10) or (100,25). We did a new simulation for N = 4, M = 1, d = 3. The result is included in the updated paper, where we observed that 3-lookahead is almost the same as 1-lookahead.
> > > > >
> > > > > 3. Since our d-lookahead policy is the "d-step lookahead oracle" that already knows the ground truths of all recovering functions, there is no sampling and/or updating posterior distributions in our d-lookahead policy. Our policy simply picks the leaf node the provides the best reward best on the ground truths of recovering functions.
> > > > >
> > > > > 4. In Alg. 1, under "If UCB", the algorithm Chooses I_t = argmax_{i\in L_d(Z_t)}.... Here, L_d(Z_t) is the set of leaf nodes in the d-lookahead tree and there are (N\choose M)^d of them. So, Alg. 1 is intractable when (N,M) = (100,10) or (100,25) and d>1.

---

### Official Review · AnonReviewer1 · 2020-10-27
**very interesting, more baselines would be welcome**

**Rating:** 7
**Confidence:** 4

**Review:**

The paper "NeurWIN: Neural Whittle Index Network for Restless Bandits via Deep RL" proposes a new RL approach for estimating the Whittle index in restless bandit problems, which allows to define effective strategies for selecting arms to activate at each round of the bandit process.

I found the paper very interesting. Maybe this was because I was not familiar with the Whittle index  and such kind of policies for restless bandits. But the paper is very didactic, I liked to discover these concepts.

More importantly, I found the proposed approach very well presented, with relevant  theoretical justification provided. The approach is elegant and looks useful from the results.

I still have some concerns however. First, one limitation of the approach is that it requires to have access to a simulator in the training phase, that strictly follows the dynamics of the real world it is designed for. It would have been nice to consider the impact of discrepancies between simulator and real-world dynamics for the setting considered. What happens if dynamics are not stationary?

Second, it would have be very useful to consider more baselines in the experiments. Authors argue that they only could applied a classical Reinforce strategy on the (4,1) setting, since the combinatorial aspect of the action and state space is problematic beyond that point.  They also claim that if this approach is not effective even in that simple setting, it is mostly due to the combinatorial dimension of the state space. Ok, but other instances of this could have been considered. At least, it would have been important to consider a policy trained with Reinforce on individual states, and then select the M best outputs at each round,  as it is done for the proposed approach ! Also, multi-agent RL approaches, such as MADDPG or Q-Mix, could have been considered to cope with the combinatorial aspect of the problem, and assess if cooperation would help improve the results.

At last, authors give conditions for the applicability of their method. It would have been useful to help the reader with some intuitions about what imply these indexability and strong indexability conditions. Are they very constraining? What kind of restless bandit settings are excluded due to these two necessary conditions ?

---

> ### Author Response · Authors · 2020-11-16
> **More results included**
>
> Thank you for the helpful and detailed review. Below, we answer the concerns raised.
>
> 1. "First, one limitation of the approach is that it requires to have access to a simulator in the training phase..."
>
> We thank the reviewer for raising this issue. To address the concern, we conduct additional experiments where the simulator is a noisy one. Specifically, the simulator has an average 5% error is its reward estimation. Due to the time constraint, we were only able to complete the experiments for the recovering bandits and the deadline scheduling. Our results show that NeurWIN still performs well despite that the simulators are noisy.
>
> 2. "Second, it would have be very useful to consider more baselines in the experiments..."
>
> We thank the reviewer for the suggestion. We implemented and tested (Fu et al. 2019), a Q-learning Whittle index algorithm, in our experiments. Results show that NeurWIN achieves superior performance.
>
> 3. "At last, authors give conditions for the applicability of their method. It would have been useful to help the reader with some intuitions about what imply these indexability and strong indexability conditions..."
>
> Both the indexability condition and the strong indexability condition basically states the following: "The net reward of activating an arm in a given state (strictly) decreases as the activation cost increases," which seems to be intuitively true for most practical problems. Indeed, many practical problems have been proven to be indexable in the literature (see below for an incomplete list). However, proving whether a problem is indexable or not is usually done on a case-by-case basis. As such, we are unable to provide a simple answer to whether a problem is strongly indexable.
>
> Borkar, Vivek S., and Sarath Pattathil. "Whittle indexability in egalitarian processor sharing systems." Annals of Operations Research, 2017.
> https://arxiv.org/pdf/1707.02440.pdf
>
> Mate, Aditya, Jackson Killian, Haifeng Xu, Andrew Perrault, and Milind Tambe. "Collapsing Bandits and Their Application to Public Health Intervention." Advances in Neural Information Processing Systems (NeurIPS), 2020.
> https://proceedings.neurips.cc/paper/2020/file/b460cf6b09878b00a3e1ad4c72344ccd-Paper.pdf
>
> Liu, Keqin, and Qing Zhao. "Indexability of restless bandit problems and optimality of whittle index for dynamic multichannel access." IEEE Transactions on Information Theory, 2010.
> https://arxiv.org/abs/0810.4658
>
> Hsu, Yu-Pin, Eytan Modiano, and Lingjie Duan. "Scheduling algorithms for minimizing age of information in wireless broadcast networks with random arrivals." IEEE Transactions on Mobile Computing, 2019.
> https://arxiv.org/abs/1712.07419
>
> Xu, Yizhen, Peng Cheng, Zhuo Chen, Ming Ding, Yonghui Li, and Branka Vucetic. "Real-Time Task Offloading for Large-Scale Mobile Edge Computing." IEEE International Conference on Acoustics, Speech and Signal Processing (ICASSP), 2020.
> https://ieeexplore.ieee.org/abstract/document/9054413
>
> Anand, Arjun, and Gustavo de Veciana. "A Whittle's index based approach for QoE optimization in wireless networks." Proceedings of the ACM on Measurement and Analysis of Computing Systems, 2018.
> http://users.ece.utexas.edu/~gustavo/papers/AnD18.pdf

---

> > ### Author Response · Authors · 2020-11-22
> > **One more update regarding strong indexability**
> >
> > We would like to provide one more update: In the latest version, we proved in Theorem 3 (Appendix C) that the deadline scheduling problem is strongly indexable. While we are unable to prove whether the other two examples (recovering bandits and wireless scheduling) used in Section 5 are strongly indexable, we would like to note that existing literature even fails to establish whether the these two examples are indexable. (For wireless scheduling, the recent paper (Aalto et al., 2015) can only prove that a relaxed version of the problem is indexable.)
> >
> > The result of Theorem 3 suggests that strong indexability may not be much more constraining than indexability.

---

### Official Review · AnonReviewer2 · 2020-10-27
**A principled approach for Whittle Index based neural network training that seems to work well in practice for restless bandit problems**

**Rating:** 7
**Confidence:** 3

**Review:**

The paper proves a simple but important result which essentially states the following: It is possible to construct an RL environment for a one arm restless bandit system, such that any neural network which achieves a certain discounted net reward has to approximate the whittle index of that arm well.

The paper then uses this observation to form exactly such an environment and train a neural-network using REINFORCE to maximize discounted reward for the environment, thus in principle achieving a good approximation of the Whittle index of the arm. The network can then be used to approximate a Whittle index based policy for restless bandit problems.

I think the idea is interesting and it seems to work well in practice on three simulated but well-studied environments. In particular in the deadline scheduling problem where the optimal strategy is known, it can be observed that this policy can match the performance of the optimal policy after several hundred training epochs.

I recommend the paper for acceptance. I would encourage the authors to provide confidence bars for their algorithm as the results are presented over 10 independent runs only. It is vital to see the variance in rewards obtained by the strategy.

---

> ### Author Response · Authors · 2020-11-16
> **Confidence intervals included**
>
> We thank the reviewer for time taken in assessing our work.
>
> We also thank the reviewer for the great advice on confidence interval. We rerun our experiments for 200 independent runs and provide confidence intervals in the figures for wireless scheduling and deadline scheduling. For recovering bandit, the problem itself is not stochastic, and hence there is only one experiment run.

---

### Official Review · AnonReviewer3 · 2020-10-28
**A more thourough comparison is needed**

**Rating:** 4
**Confidence:** 4

**Review:**

This paper considers the problem of learning how to control restless bandits.  When all parameters of the system are known, Whittle index policy usually offers a good performance. The main contribution of the paper is to propose an algorithm, NeurWIN, that uses a neural network architecture to learn the Whittle indices. Most of the paper is devoted to the description and the derivation of the algorithm. At the end of the paper, the authors present four illustrations of how the algorithm works. The algorithm is compared to an index-based policy and an (old?) RL algorithm REINFORCE for the small systems. The learning behavior of NeurWIN is very good in all tested cases.

The paper does not contain theoretical result. It is mostly about presenting an algorithm and performing numerical experiments to demonstrate that it works. Yet, there is no comparison with (reasonable) related work. In the numerical part, the only algorithm to which the authors compare their algorithm is REINFORCE algororithm of Williams 92, that is clearly non-adapted to learning whittle index (because of the large state-space). As acknowledged by the authors, there are a number of recent work on learning Whittle index (last paragraph of Section 2): I do not understand why these solutions are not implemented. Also, I am not convinced that the algorithm of (Avrachenkov,Borkar,2019) is not applicable to the current context.

The paper is a bit sloppy and not very precise. For instance:
- The authors write several times "Finding Whittle index is typically intractable" without proof or precise reference.  I doubt that this statement is true.
- page 4, top: the definition of strong indexability needs some clarifications: how strong is the assumption? Do the presented example satisfy this assumption? And if not, is it important?
- page 4: is it really the problem statement? Then why are the simulation results not showing this?
- page 7 (end): "The size of the state-space is 10^{12} : given the appendix, this seems largely exaggerated (the order seems closer to 10^6 if one discretize byte per byte). Moreover, discretizing at the kB level would lead to about 10^3 states in which case Whittle index would be applicable.
- page 8 (beginning): "REINFORCE [...] improves a litle" -> this seems wrong.
- page 8, Figure 4(a): how can NeurWIN perform better than Whittle index? I thought that NeurWIN was learning those indices.

The paper would be much stronger, if it would contain either some theoretical guarantee, or a more thourough comparison of performance with related work.

---

> ### Author Response · Authors · 2020-11-12
> **Response to comments and suggestions**
>
> We thank reviewer #3 for the constructive and detailed feedback.
> We discuss each point the reviewer gave below.
>
> 1. Regarding the comparison with recent work:
>
> We thank the reviewer for raising this issue. We will implement (Fu et al. 2019), a Q-learning Whittle index algorithm, and compare its performance with NeurWIN. We would like to point out that (Fu et al. 2019) is a tabular method that may not scale well. Further, the paper itself shows that its algorithm does not converge to the Whittle index. Still, as it is the closest recent work to our NeurWIN, we will implement and test it.
>
> For (Avrachenkov,Borkar 2019), the proposed algorithm is only applicable when the state of an arm is a scalar. In this paper, we consider that the state of an arm can be a vector. Indeed, the applications in Section 5.3 and 5.4 both have vector states. Hence, (Avrachenkov,Borkar 2019) is not applicable.
>
> 2. Regarding the hardness of finding the Whittle index:
>
> To the best of our knowledge, there is no tractable solutions for finding Whittle indices. The standard approach, as described in Section III.B in (Yu et al., 2018), finds Whittle indices by solving uncountably infinitely many linear programming problems, one for each activation cost \lambda in (-\infty, +\infty). This approach is clearly not tractable. While we are not aware of any hardness result of finding the Whittle index, we would like the reviewer to consider the following two points:
>
> A. Most existing work on Whittle index is only able to find the Whittle index for some special cases.
> B. In (Aalto et al., 2015), which we discussed in Section 5.3, the authors were only able to find the Whittle index under some relaxations.
>
> If finding Whittle indices were tractable, researchers wouldn't have resorted to studying special cases and/or using relaxations.
>
> 3. "page 4, top: the definition of strong indexability needs some clarifications: how strong is the assumption? Do the presented example satisfy this assumption? And if not, is it important?"
>
> We need the strong indexability condition to prove theorem 2. However, NeurWIN can still be applied if this condition is not met.
>
> We would also like to point out that the strong indexability condition basically states the following: "The net reward of activating an arm in a given state strictly decreases as the activation cost increases," which seems to be intuitively true for most practical problems. In the literature, proving whether a problem is indexable or not is usually done on a case-by-case basis. Proving whether a problem is strongly indexable is beyond the scope of this work.
>
> 4. "page 4: is it really the problem statement? Then why are the simulation results not showing this?"
>
> The problem statement is to find neural networks that are \gamma-Whittle-accurate. In the first two applications that we evaluate (Sections 5.2 and 5.3), there are no known ground truths about the Whittle indices. Hence, we are unable to evaluate whether NeurWIN is Whittle-accurate. We can only evaluate NeurWIN by its resulting rewards. As the rewards are higher than state-of-the-art policies, it appears that NeurWIN must produce accurate predictions of Whittle indices.
>
> For the application in Section 5.4, we do have the ground truths and we will include the comparison with ground truths in the paper. What we observe is that NeurWIN is accurate when T is small, but is less accurate when T is large. This is reasonable. In Section 5.4, T is the time to deadline. When the time to deadline is large, the actions have little impact on the resulting performance. However, when T is small, making the wrong action can result in big penalty. The fact that NeurWIN is more accurate when T is small suggests that NeurWIN focuses more on learning the Whittle indices for states with higher importance.
>
> 5. "page 7 (end): "The size of the state-space is 10^{12} : given the appendix, this seems largely exaggerated..."
>
> We thank the reviewer for spotting the typo. The size of the state-space is 2*10^6. The size of the file ranges from 1 Byte to 10^6 Bytes. The channel can be either good or bad. We will correct the typo.
>
> 6. "page 8 (beginning): "REINFORCE [...] improves a litle" -> this seems wrong."
>
> We thank the reviewer for spotting the typo. We meant to say "REINFORCE...improves little," meaning that it does not improve. We will correct the typo.
>
> 7. "page 8, Figure 4(a): how can NeurWIN perform better than Whittle index? "
>
> As mentioned in page 1 end of paragraph 2,  Whittle index policy is asymptotically optimal in many settings. Here, "asymptotically optimal" means that it converges to the optimal policy when the number of arms becomes large. For a small number of arms (e.g. N = 4), there may exist a family of control policies that perform better than the Whittle index-based policy.
> For NeurWIN with a small set of arms N, we observed that the learned index representations converged to one such control hypothesis.

---

> > ### Comment · AnonReviewer3 · 2020-11-16
> > **Detailed comments**
> >
> > 1. "We will implement (Fu et al. 2019) [...] For (Avrachenkov,Borkar 2019), the proposed algorithm is only applicable when the state of an arm is a scalar."
> > Having more comparison would strengthen the paper. However, I am not convinced that (Avrachenkov,Borkar 2019) is only applicable to scalar arms (and in fact, what makes an arm "scalar": if the state space is finite, one can always reorder the state space on a line {1\dots S}, no?
> >
> > 2. The authors should look at the series of paper from Nino-Mora, like "Dynamic priority allocation via restless bandit marginal productivity indices". What is difficult in general is to prove indexability and to obtain closed form results. IMHO, this is why researchers restrict themselves to particular cases.
> >
> > 3. I take the point that NeurWIN can still be applicable even if the problem is not strongly indexable. Yet, when a definition like this is given it might be worth commenting on whether the examples used satisfy this condition.
> >
> > 4. My point here is that the paper uses NeurWIN to design efficient policies by learning Whittle index. I believe that the fact that NeurWIN learns the index is a means rather than a goal.
> >
> > The rest of the answer are convincing.

---

> > > ### Author Response · Authors · 2020-11-22
> > > **Responses**
> > >
> > > 1. Regarding (Avrachenkov,Borkar 2019): We would like to provide a more detailed discussion about this paper. As the beginning of Section IV of (Avrachenkov,Borkar 2019) shows, this paper assumes that V(x), a function related to the Whittle index, as a linear combination of a set of predetermined feature functions, \phi_m(x). The learning algorithm aims to find the weight r(m) of each feature function. There are two issues about this approach:
> > >
> > > A. This work requires choosing an appropriate set of feature functions. It cannot approximate V(x) well unless V(x) can be well-approximated by the chosen set of feature functions.
> > >
> > > B. This approach implicitly assumes that V(x) is smooth in x. Thus, when applying the reordering technique suggested by the reviewer, the ordering of states can have significant impact on the performance. Intuitively, the ordering should be chosen such that adjacent states have similar Whittle indices. Of course, without knowing the Whittle index in advance, it is impossible to find the appropriate ordering.
> > >
> > > Due to the above issues, we do not find (Avrachenkov,Borkar 2019) to be a good baseline. Instead, we implemented and tested (Fu et al. 2019), whose model is consistent with ours. As shown in the latest version, our policy significantly outperforms (Fu et al. 2019).
> > >
> > > 2. Regarding the work from Nino-Mora: We thank the reviewer for suggesting this reference. We read the paper carefully and below is our summary:
> > >
> > > In Section 2.2, the paper described a generic way to find the Whittle index. The approach is to first construct Fig. 1, which requires finding f^S and g^S for all subsets S of states. Hence, the complexity is at least exponential to the number of states. We believe this section serves as a justification to our statement "Finding Whittle index is typically intractable". We included the following footnote to justify the statement:
> > >
> > > \cite{nino2007dynamic} described a generic approach for finding the Whittle index. The complexity of this approach is at least exponential to the number of states.
> > >
> > > In Section 2.3, the paper described another way to find the Whittle index with lower complexity. This approach is only applicable when the arm is PCL-indexable.
> > >
> > > 3. Regarding strong indexability: We thank the reviewer for the comment. We included a new theorem that proved the deadline scheduling problem is strongly indexable. Please see Theorem 3 in the appendix. We are unable to prove whether the other two examples (recovering bandits and wireless scheduling) used in Section 5 are strongly indexable. However, we would like to note that existing literature even fails to establish whether the first two examples are indexable. (For wireless scheduling, the recent paper (Aalto et al., 2015) can only prove that a relaxed version of the problem is indexable.)
> > >
> > > 4. Regarding "goal" vs "means": We thank the reviewer for the comment. We revised the problem statement to highlight that finding Whittle index is the means, and the goal is to solve the restless bandit problem. The statement now reads:
> > >
> > > Our goal is to derive low-complexity index algorithms for restless bandit problems by training a neural network that approximates the Whittle index of each restless arm using its simulator.

---

### Author Response · Authors · 2020-11-16
**Paper updated**

We thank the reviewers for the great comments and suggestions. We have incorporated them and updated the paper. In particular, we include the following new results:

1. We implemented and tested the QWIC algorithm (Fu et al. 2019). This is an algorithm aiming to find Whittle index through Q-learning. It appears to be the work most related to ours. Results show that NeurWIN performs much better than QWIC.

2. We considered the possibility that the simulator is not perfectly precise in simulating the environment. In particular, we conducted training when the simulator has an average 5% error in its estimated rewards. Results show that, despite being trained on a noisy simulator, NeurWIN still achieves good performance.

3. For problems that are stochastic, we conducted 200 independent runs and reported the confidence intervals.

4. We revised multiple parts of the paper to address some misunderstandings.

---

### Decision · Program_Chairs · 2021-01-07
**Final Decision**

**Decision:**

Reject

**Comment:**

This paper approximates the Whittle index in restless bandits using a neural network. Finding the Whittle index is a difficult problem and all reviewers agreed on this. Nevertheless, the scores of this paper are split between 2x 4 and 2x 7, essentially along the line of whether this paper is too preliminary to be accepted. Therefore, I read the paper and propose a rejection.

The reason is that the paper lacks rigor, which was brought up by the two reviewers who suggested rejections. For instance, in the last line of Algorithm 1, it is not clear what kind of a gradient is computed. The reason is that \bar{G}_b is not a proper baseline, as it depends on the future actions of the bandit policy in any given round. I suggest that the authors look at recent papers on meta-learning of bandit policies by policy gradients,

https://papers.nips.cc/paper/2020/hash/171ae1bbb81475eb96287dd78565b38b-Abstract.html

https://arxiv.org/abs/2006.16507

This is the level of rigor that I would expect from this paper, to make sure that the gradients are correct.